# The Damage Performance of Uncarbonated Limestone Cement Pastes Partially Exposed to Na_2_SO_4_ Solution

**DOI:** 10.3390/ma15238351

**Published:** 2022-11-24

**Authors:** Yu Cui, Min Pei, Ju Huang, Wei Hou, Zanqun Liu

**Affiliations:** 1School of Civil Engineering, Central South University, Changsha 410075, China; 2Hunan CJS Technologies Co., Ltd., Changsha 410000, China; 3School of Materials and Chemical Engineering, Hunan Institute of Technology, Hengyang 421000, China

**Keywords:** sulfate attack, carbonation, partial immersion, cement paste, pore structure

## Abstract

Pore structure and composition of cement paste are the main two factors in controlling the sulfate attack on concrete, but the influence of carbonization on pore structure and composition is often ignored in sulfate attack. Therefore, will the damage performance of concrete partially exposed to sulfate solution be different avoiding the alterations of pore structure and composition due to carbonation? In this paper, the cement pastes were partially immersed in 5 wt. % sodium sulfate solution, with N_2_ as protective gas to avoid carbonation (20 ± 1°C, RH 65 ± 5%). Pore structures of cements were changed by introducing different contents of limestone powders (0 wt. %, 10 wt. %, 20 wt. %, and 30 wt. %) into cement pastes. The damage performance of the specimens was studied by ^1^H NMR, XRD and SEM. The results showed that the immersion zone of pure cement paste under N_2_ atmosphere remained intact while serious damage occurred in the evaporation zone. However, the damage of cement + limestone powders pastes appeared in the immersion zone rather than in the evaporation zone and cement pastes containing more limestone were more severely damaged. Compositional analysis suggested that the damage of the evaporation zone or the immersion zone was solely caused by chemical attack where substantial amount of gypsums and ettringites were filled in the pore volumes. Introduction of limestone powders led to the increase of the pore sizes and porosity of cement pastes, causing the damage occurred in the immersion zone not in the evaporation zone.

## 1. Introduction

When concretes are partially exposed to groundwater and sulfate-bearing soil, damage is usually produced near the evaporation zone while the immersed part directly buried in groundwater or soil remains intact [1,2]. Studies have shown that the damage in the evaporation zone of concrete is similar to the damage in stone caused by salt crystallization, i.e., the physical sulfate attack [3]. In fact, the great difference exists between cementitious materials and stones: barely no sulfation reactions take place in stones but in concretes [4,5]. 

To date, even though many research studies have been reported on the damage mechanisms of cement-based materials that are partially immersed in sulfate solutions, controversies still linger on the mechanism of damage. Chen [6] and Xiong [7] found that both chemical reaction and salt crystallization took place in the evaporation zone of cement-based materials, and the latter was the driving mechanism for cracking in the evaporation zone. Huang [8] pointed out that chemical attack was the direct cause in the evaporation zone of the specimens. Clearly, according to the salt crystallization pressure theory [9,10] (salt crystals growing from a supersaturated solution and exerting destructive pressure on the pore wall), the precondition for physical salt crystallization damage to occur is that the salt solution must be supersaturated. Capillary adsorption leads to upward migration of sulfate ions and thus an increase in the concentration of sulfate ions in the upper part due to water evaporation. However, cement hydration products (e.g., Ca(OH)_2_) prefer to react with sulfate ions to produce expansive products, such as ettringites and gypsums [11,12,13], which consume sulfate ions, making it difficult for the salt solution to reach supersaturation before chemical erosion damage occurs. The pore structure of the specimen influences the height of capillary rise, the rate of water evaporation in the upper part of the specimen, and permeability of sulfate ions [3,14,15]. These affect the physical salt crystallization and chemical rection in the specimens. Therefore, the destruction of specimens partially exposed to the sulfate solution greatly depends on the pore structures.

Many experimental studies have ignored the effect of carbonation on physical salt attack [16]. Carbonation alters the pore structure and phase composition of cement specimens [17,18] and significantly affects the damage to the evaporated zone of concrete under partially immersed in sulfate solutions. By preforming accelerated carbonation and then partially immersed the specimens in Na_2_SO_4_ or MgSO_4_ solutions, Liu [19,20] and Yoshida [2] found that salt crystallization was the direct cause of damage to the specimens and that the greater the carbonation depth, the more serious damage of salt crystallization. Moreover, compared with the references, the mass loss rate of the accelerated carbonized specimen is the largest due to physical salt attack. Hence, to understand the effect of pore structure on the destruction mechanism of Portland cement partially exposed to sulfate environments, carbonation should be avoided. The limestone powder is often used as an inert supplementary material to adjust the pore structure of components. When limestone powders are dosed at less than 10%, it can refine the pore structure [21], whereas excessive limestone powders lead to coarsening of the pore structure and with the increase of porosity [22,23].

The aim of this study is to illuminate the effect on the mechanism of destruction mechanism of partially exposure to specimens with the pore structure changes. Therefore, in this study, the composition and microstructure of the cement paste specimens were adjusted by contents of limestone addition. Subsequently, the specimens were partially immersed in 5 wt. % Na_2_SO_4_ solution, and the specimens were placed in N_2_ environment. Additionally, ^1^H Nuclear Magnetic Resonance (^1^H NMR), X-ray Diffraction (XRD) and Scanning Electron Microscopy (SEM) microscopic analyses were used to study the mechanism of damage. This study provides evidence that aids understanding the pore structures effect on sulfate attack. 

## 2. Experimental Procedures

### 2.1. Materials

P I 42.5 cement and high purity limestone powder (CaCO_3_ content up to 98%) produced by Hubei Jingmen Gao Xu Chemical Co., Ltd. Were used in this study. The chemical compositions of cement and limestone power are shown in Table 1. The d_50_ of cement and limestone powder determined by laser particle size determination are 13.096 μm and 14.426 μm, respectively. Na_2_SO_4_, MgCl_2_·6H_2_O, and Ca(OH)_2_ were analytical reagent. 

### 2.2. Specimens Preparation and Exposure Conditions

The fresh pastes were mixed according to the mixing proportion in Table 2 and cast in 7 mm × 40 mm× 160 mm molds. The pastes were covered with plastic film to avoid water evaporation and cured at a constant temperature room (20 ± 1 °C). After 24 h, specimens were demolded and cured in a saturated Ca(OH)_2_ solution at 20 ± 1 °C for 56 days. For comparison, some specimens were taken out, dried on the surface, and put into a vacuum desiccator containing silica gel desiccant for 7 days. The remaining specimens were maintained in saturated Ca(OH)_2_ solution as a reference. The dried specimens were partially immersed in a 5% Na_2_SO_4_ solution for 80 days (the solution was replaced once 15 days to keep the concentration constant [24]) with an immersion depth of 5 cm. During this period, some partially immersed specimens were placed in an air environment while the remaining partially immersed specimens were placed in a sealed environment chamber with ≥95% nitrogen gas. The temperature and humidity were maintained at 20 ± 1 °C and 65 ± 5% in both environments.

### 2.3. Test Setup

The experimental setup mainly consisted of a sealed environment box, a specimen immersion box, a solution box, a nitrogen concentration tester, and a liquid nitrogen tank, as shown in Figure 1. The specimen box with a height of 40 mm contains a 5% Na_2_SO_4_ solution. When the specimens were partially immersed in this box, the nitrogen gas was provided as protective gas in sealed chamber. The humidity in the sealed environment box was controlled within 65 ± 5% using saturated MgCl_2_ solution in the solution box.

### 2.4. ^1^H NMR, XRD and SEM Analysis

The pore structures of specimens were determined by ^1^H NMR. After certain curing ages, the specimens were removed from the solution, and the surfaces of the specimens were cleaned with a brush. The evaporation and immersion zone of the specimens were cut according to the position shown in Figure 2a. The samples were saturated with water in vacuum to a constant weight [25] prior to test. A MicroMR12-025 manufactured by Newmark Electronic Technology Ltd., California, USA was used, with a resonance frequency of 11.793 MHz, a magnet temperature control of 35.00 ± 0.02 °C, and a probe coil diameter of 25 mm. 

XRD tests were performed for compositional analysis. When the experiment was conducted at 50 and 80 days, the samples were collected from the different parts of specimens (see Figure 2a) and then were sliced with a thickness of ca. 1 mm (3 mm in total) in evaporation zone (see Figure 2b), to identify any corrosion performances in different layers of the evaporation zone. The powder sample were finely ground with isopropanol in an agate mortar and dried in a vacuum desiccator for 7 d. Then, the samples were mixed with 20 wt. % ZnO (an internal standard). The XRD patterns of samples were collected using an Automated D/max-III X-ray diffractometer from Rigaku, Tokyo, Japan with a Cu-Kα target, an acceleration voltage of 40 kV, and a current of 40 mA. The scan ranged from 5° to 65°. The scanning step and rate is 0.02° and 2°/min, respectively. 

After the immersion test, the microstructures of deteriorated specimens were analyzed by SEM. The deteriorated portion was dried in a vacuum desiccator for 7 d. Afterward, the sample was then placed in a vacuum coater for gold spraying. A Nova Nano-SEM 230 field emission SEM from FEI Electron Optics B.V., Oregon, USA was used to observe the microscopic morphology of the samples in secondary electron mode with an acceleration voltage of 18 kV. 

## 3. Results and Discussion 

### 3.1. Damage of Specimens

Figure 3a shows pictures of specimens after 80 days of exposure to 5% Na_2_SO_4_ solution in air. The surface of the evaporation zone of all specimens was covered with considerable whitish crystals. The formation of whitish crystals on the evaporation zone surface is a clear indicator for salt crystallization. The addition of limestone the LS20 specimen fractured in the evaporation zone (after 50 days of partial immersion, the LS30 specimen had already fractured in the evaporation zone). The evaporation zone of all specimens had different degrees of deterioration. The degree of damage in the evaporation zone was more serious in all specimens than in the immersion zone. 

However, the damage characteristics of the specimens in N_2_ environment were different from those in air. After 80 days of partial immersion (Figure 3b), only a small amount of whitish substance appeared on the surface of the evaporation zone of all specimens. For PC specimens, the evaporation zone cracked, and the immersion zone remained intact (Figure 3c), but LS specimens were the opposite. This suggests that. For specimens containing limestone in nitrogen atmosphere, the location of damage changed depending on the amount of limestone incorporated into the specimens. The specimens containing 30 wt. % limestone showed negligible deterioration in the evaporation zone, while the damage in the immersion zone was obvious. Expansion, cracking, and spalling are found on the pastes surface (Figure 3d). This is most likely due to the fact that the high dosing of limestone powder changed the microstructure of the specimens. 

### 3.2. Pore Structure Analysis

The pores provided volumes for expansive crystals growth during sulfate attack and channels for ions diffusion, and therefore pore structure is important target for analysis on damage from sulfate attack. Here, the pore size distributions of the specimens before sulfate attack are shown in Figure 4. The pore sizes are classified into three ranges: <100 nm, 100–200 nm, and >200 nm.

The total porosity of the specimens gradually increased with increasing in limestone addition. As the limestone addition increased from 10 wt. % to 30 wt. %, the total porosity was increased by 0.4% (LS10), 3.7% (LS20) and 6.6% (LS30), respectively. It was found that the pore radius of specimen containing limestone were coarsened. Specifically, the percentage of pores <100 nm in LS specimens decreased slightly, while the proportion of larger pores (>100 nm) increased by 2.7% (LS10), 6.1% (LS20), and 10.4% (LS30), respectively. It has shown [26] that the addition of 8 wt.% coarse limestone powder (>10 μm) to the cement was able to significantly increase the >100 nm pores in the cement paste and increased the total porosity of the cement paste. This may be related to the fact that introducing 30 % inert limestone powders produced significant “dilution effect”. This resulted in a reduced cement content and thus increased in the effective w/c of the test mix, leading to the formation of coarser pores. According to the capillary adsorption principle [27] (Equation (1)), the height of capillary adsorption rise is determined by the pore radius. The smaller the pore size, the higher the height of capillary adsorption. With the increase of limestone powder addition, the pores in the specimen are coarsened, and therefore, the capillary rise height in the evaporation zone is significantly reduced. This might be responsible for much slight damage of LS samples in the evaporation zone. On the other hand, the larger pores also facilitate the SO_4_^2-^ diffusion [15] (Equation (2)), which causes more serious sulfate attack in the immersion zone of LS specimens than in the evaporation zone.
(1)$$h=\frac{2\gamma \cos\theta }{\Delta \rho gR}$$
where *h* is the liquid rise height, m. Δ*ρ* is the liquid-gas phase density difference, kg/m^3^. *g* is the acceleration of gravity, 9.8 m/s^2^. *R* is the capillary radius, m. *γ* is the liquid surface tension, N/m. *θ* is the contact angle, °.
(2)$$D^{*}=D\frac{\varphi }{\tau }$$
where *D** is the effective diffusion coefficient of sulfate ions, m^2^/s. *D* is the diffusion coefficient of sulfate ions in solution, m^2^/s. *φ* is the concrete porosity, %. *τ* is the concrete curvature (a constant in this study).

After immersion test, the porosity in different parts of the specimens of both PC and LS30 specimens were investigated, as shown in Figure 5. 

The porosity in the evaporation and immersion zones was lower than that of the reference sample (that immersed in saturated Ca(OH)_2_ solution for 136 days). The total pore volume was ascribed to the consequence from ‘‘filling effect” [24]. Specifically, the porosity in the evaporation zone of the PC specimen was reduced by 5.5%, as a result of reduction in pores with diameter of <100 nm (−3.77%) and 100–200 nm (−1.77%). While the pore size in the immersion zone was only reduced by 0.4%. As pore size diminishing is related to the filling of expansive product formed in sulfate attack, this indicates that sulfate attack reactions in PC specimens might mainly take places in the evaporation zone. This explains that in Figure 3c, the damage occurred in the evaporation zone while the immersion zone remained intact.

However, the porosity reduction in limestone cement is quite opposite. The total porosity in the evaporation zone of LS30 specimens was only decreased by 1.3% while that in the immersion zones was reduced by 3.7%. This suggests that amounts number of large pores facilitated the diffusion of sulfate ions into the immersion zone to sulfate attack reactions, and expansive reaction products mainly occupied the pore volumes in the immersion zone. More specifically, the reduction in porosity in the immersion zone of L30 samples is attributed to the significant decrease (by 3.4%) in the small pores (<200 nm). The result obtained is similar to that seen by Ikumi et al. [28]. This demonstrates the small pores are preferred sites for chemical reactions to happen. Moreover, according to theory of crystallization pressure, when the crystals filled in relatively limited space, considerable pressure may cause pore or even specimens cracking [29].

### 3.3. Compositional Analysis

The damage of partially immersed specimens in sodium sulfate solution was found after 80 days. To understand the composition evolutions in the specimens before and after the damage, samples after 50 days (undamaged) and 80 days (damaged) were taken for analysis. Figure 6 shows the XRD patterns of the undamaged specimens after 50 days in sodium sulfate solution. 

The powders from evaporation zone and the immersion zone in the specimens and the powders from reference sample (that immersed in saturated Ca(OH)_2_ for 106 days) were sampled and analyzed by XRD test. Compared with the reference sample free of sulfate attack and carbonation, the increase in the diffraction peaks intensity of ettringite and gypsum in the PC specimens subjected to 50 days of sulfate attack is noticeable. This is a clear indicator for the massive formation of ettringite and gypsum crystals during sulfate attack in both evaporation zone and immersion zone. However, it seems that more expansive products were formed in the evaporation zone. This may be related to the capillary adsorption action provided by the pores with small radius, causing more sulfate ions with Ca(OH)_2_ reacted. LS cements also see an obvious increase in the content of ettringite and gypsum crystals in both evaporation zone and immersion zone, whereas the content of formed products from sulfation reactions is much higher in the immersion zone than that in the evaporation zone. This becomes more pronounced as the increase of limestone addition. This also shows that the foreign sulfate ions are preferred to react with hydrates in the immersion zone of LS samples due to larger pores radius enhance sulfate ions diffusion. However, it should be noted, at least at this time, no sodium sulfate crystals were detected in any parts of specimens, which suggests that salt crystallization did not occur along with sulfate attack in all the specimens. Even though considerable expansive products had formed after 50 days, no obvious damage was observed on the surface of specimens. 

Distinct damage was found only after 80 days in evaporation zone of PC samples but in immersion zone of LS samples. A few white crystals were also found on the surface of PC samples, and it is thus suspicious for the salt crystallization in the evaporation zone. Therefore, the samples in the evaporation zone were sliced with a thickness of ca. 1 mm (3 slices in total) and compositions in each layer were determined by XRD (shown in Figure 7). 

However, in Figure 7a, no sodium sulfate crystals were found, even at the outer layer (E1) of the evaporation zone. This confirms that salt crystallization is not the reason for the damage in this part. Instead, the prominent increase in the ettringite and gypsum crystals in each layer was noticed. There is no great difference in the ettringite and gypsum contents in each layer. In the immersion zone, however, the content of ettringite and gypsum is 5.4 wt. % and 4.5 wt. %, which is much less than that in the evaporation zone (11.5 wt. % and 5.5 wt. %, respectively), as shown as Figure 8. This might be the reason why the immersion zone of PC samples remained intact at the end of experiment. The evidence demonstrates the chemical reactions should be responsible for the damage of PC samples in the evaporation zone.

In contrast, for LS specimens, diffraction peaks of ettringite and gypsum become more intensive in the immersion zone rather than in the evaporation zone (Figure 7b–d). The quantitative analysis of ettringite and gypsum were carried out and the results are shown in Figure 8. 

Clearly, the content of ettringite in the immersion zone of LS30 reached 11.4 wt. % while in the outer layer (E1) and the inner layer (E3) of the evaporation zone, it was only 7.8 wt. % and 2.4 wt. %. In the E2 and E3 layer, the content of gypsum is too low to be detected. While the weakening of capillary adsorption reduces the concentration of sulfate ions in the evaporation zone, slowing down the degree of erosion, under the influence of water evaporation, the E1 sulfate concentration in the evaporation zone of the samples is higher than the E2 and E3 [27], so that more chemical products are produced. The results illustrate that the immersion zone of LS30 samples was more severely attacked by the sulfate ions than the evaporation zone. Consequently, damage on LS samples was found at the immersion zone instead of evaporation zone. The contents of ettringite and gypsum in the immersion zone decreased when less limestone was incorporated. The content of these expansive products reached 12.9% (LS10), 15.5% (LS20), and 15.7% (LS30), respectively. This explains why LS30 and LS20 specimens were more severely damaged than LS10 (undamaged). The reason behind may be that increase in the limestone addition leads to the larger porosity that favors sulfate ions diffusion [15], as discussed above.

### 3.4. Microstructure Analysis

Jiang [1] claimed that the specimens partially exposed to salt solutions, the absence of the characteristic peaks of sodium sulfate crystals in XRD was subjected to the crystals being too little, but the little crystals could be detected by SEM-EDS. Therefore, the morphology of fragments from the surface in Figure 3c,d were observed using SEM and the images are shown in Figure 9.

A large number of needlelike ettringite and massive gypsum were present on the damaged surface of the evaporation zone of the PC specimen. Ettringite grew gradually from the circular pore boundary and slowly filled the pores, and many cracks were found around the gypsum and ettringite. Liu [30] claimed that expansive reaction products in the pores generated crystallization pressure easily, which caused the cracks of pores. The sodium sulfate crystals were not observed, indicating that the cracking damage at the edges of the evaporation zone of the PC specimens was not related to the crystallization of sodium sulfate crystals. Instead, the formation of expansive crystals from the chemical reactions between the foreign sulfate ions and hydrates is the main reason for such damage. Ettringite and gypsum crystals were present in the LS30 immersion zone fragments and numerous cracks were produced. However, gypsum crystals in the LS30 submerged zone were much larger in size. According to previous studies [31,32], the cement-based materials containing large amounts of limestone powder are often subjected to volume expansion, cracking and flaking in a sulfate environment. This is mainly related to the significant increase in porosity that promotes the sulfate ions invasion into the cement specimen, which then react with hydrates to form massive expansive products. Müllauer [33] found that ettringite filled in small pores generated a stress, which was responsible for expansion and damage. Even though ettringite and gypsum formation in larger pores also takes place, its contribution to expansion and damage is negligible. Therefore, as mentioned above for XRD and SEM, these results also indicated that the chemical reaction produces a large amount of ettringite filling the small pores and generating expansion stresses, which leaded to cracking of the specimen. 

## 4. Conclusions

In this study, specimens with different pore structures were subjected to the sulfate attack by partial immersion in sodium sulfate solutions. The specimens were protected in the nitrogen atmosphere to avoid the pore size and composition evolution resulting from carbonation. In this way, the effect pore structure of cement on sulfate attack mechanism were discussed and the following conclusions can be draw from this paper.

(1)The large quantity of small pores (<100 nm) in PC specimens caused strong capillary adsorption that favors the migration of sulfate ions to the evaporation zone. The reactions between these ions and hydrates led to the massive formation of expansive products (ettringite and gypsum) that were filled in the pores and caused cracking of the specimen, whereas the damage in the immersion zone was relatively slight.(2)Limestone addition significantly increased the larger pores (>100 nm) while decreased the finer pores (<100 nm) in specimens, which resulted in a gradual coarsening of the pore structure and an increase in porosity. As such, the capillary absorption effect in limestone-cements was weakened, and, therefore, sulfate attack in the evaporation zone was alleviated. However, the larger pores promoted diffusion of sulfate ions and thus serious damage chemical attack was found in the immersion zone.

Therefore, in the conditions free of carbonation, it is not the salt crystallization but the chemical reactions between sulfate ions and hydrates that should be responsible for the damage of cements under sulfate attack. As the increase in the large pores in the cements, the damaged part shifts from the evaporation zone to the immersion zone.

## Figures and Tables

**Figure 1 materials-15-08351-f001:**
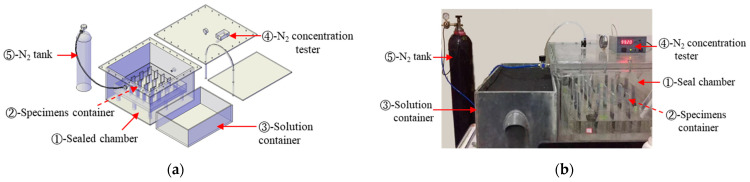
Experimental setup. (**a**) Schematic diagram (**b**) Experimental setup.

**Figure 2 materials-15-08351-f002:**
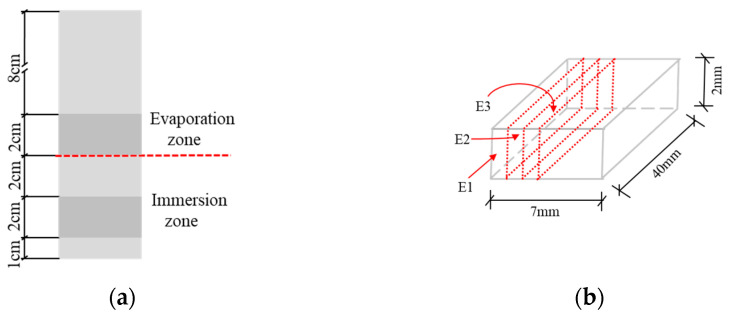
Sampling method of the specimen. (**a**) Different zones of the specimen. (**b**) Sampling in evaporation zone.

**Figure 3 materials-15-08351-f003:**
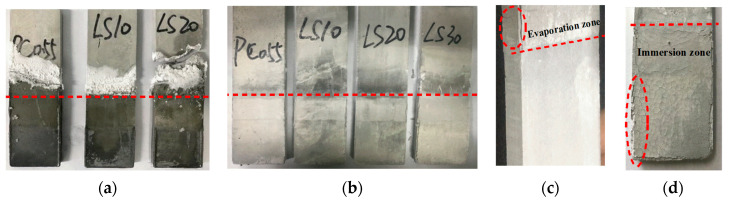
Damage characteristics of specimens partially immersed for 80 days, under different environments. (**a**) In air, (**b**) in N_2_, (**c**) PC in N_2_, and (**d**) LS30 in N_2_.

**Figure 4 materials-15-08351-f004:**
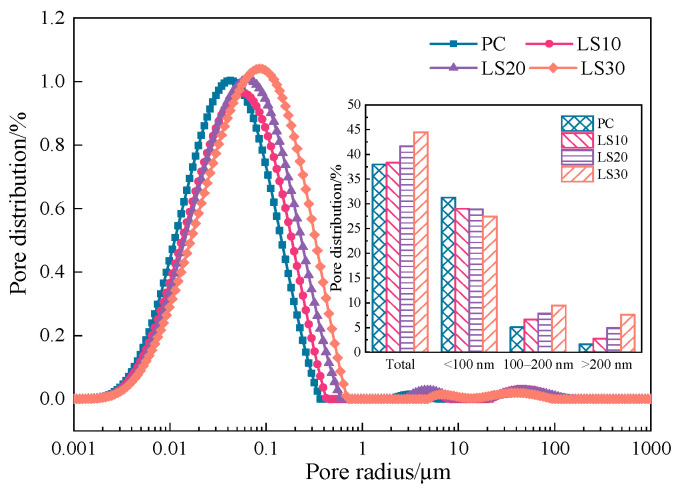
Pore size distribution in specimens cured for 56 days.

**Figure 5 materials-15-08351-f005:**
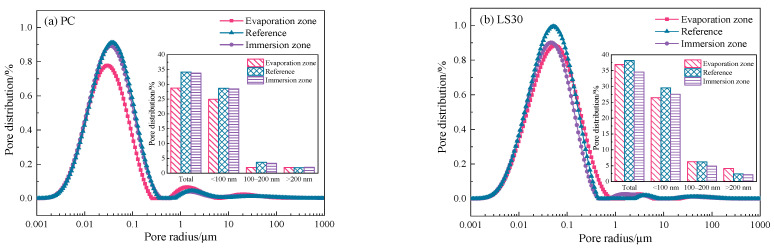
Pore size distribution in evaporation zone, immersion zone, and reference of partially immersed for 80 days specimens.

**Figure 6 materials-15-08351-f006:**
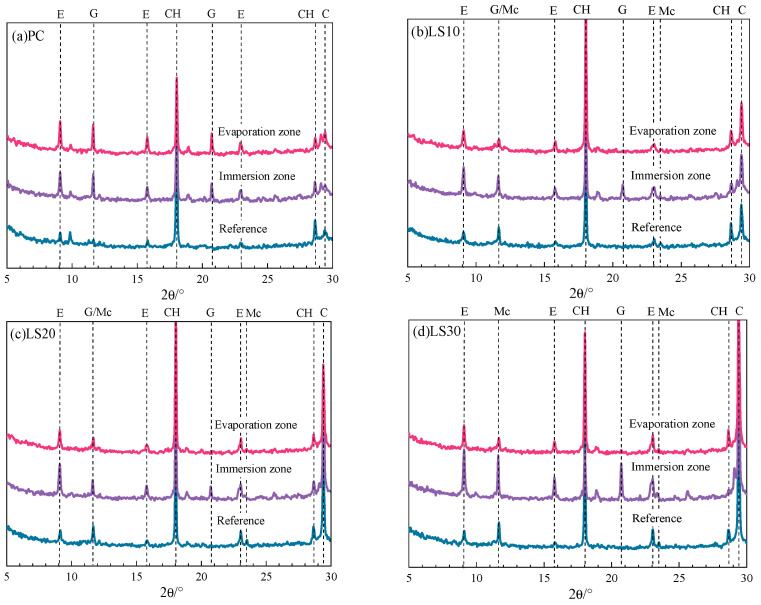
XRD pattern of specimens with different limestone content after partially immersed specimens for 50 days. (E—ettringite; G—gypsum; Mc—monocarboaluminate; CH—portlandite).

**Figure 7 materials-15-08351-f007:**
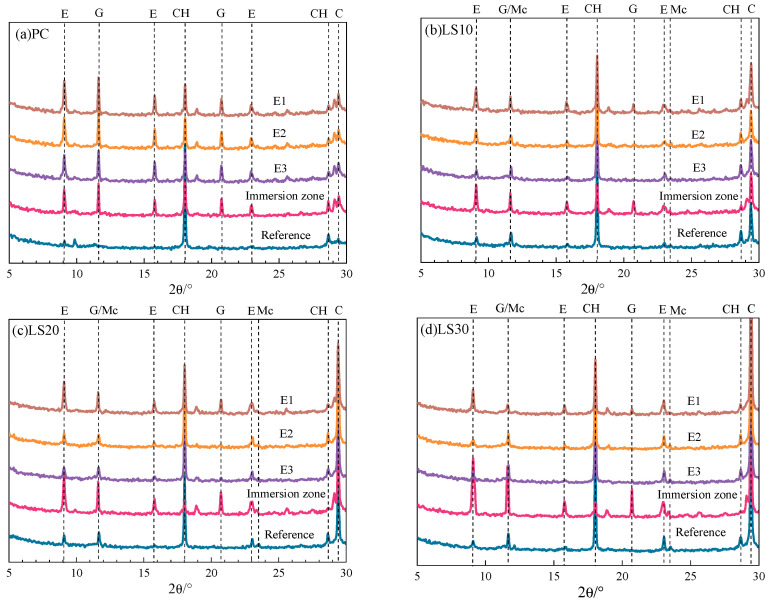
XRD pattern of specimens with different limestone content after partially immersed specimens for 80 days. (E—ettringite; G—gypsum; Mc—monocarboaluminate; CH—portlandite).

**Figure 8 materials-15-08351-f008:**
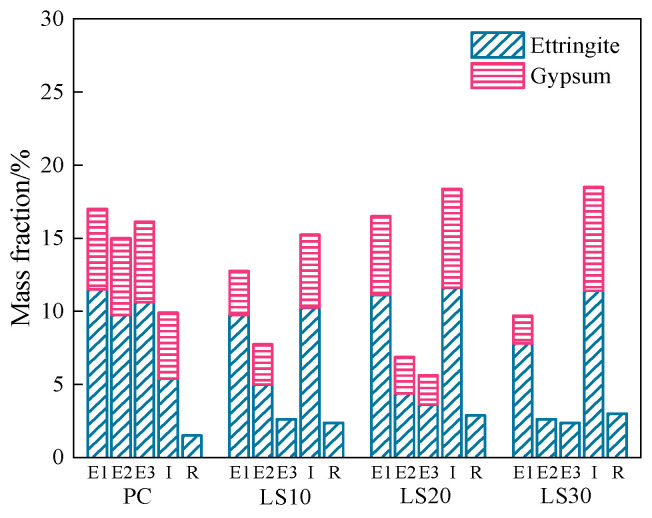
Mass fraction of main chemical products in each part of the specimens with different limestone content after partially immersed specimens for 80 days.

**Figure 9 materials-15-08351-f009:**
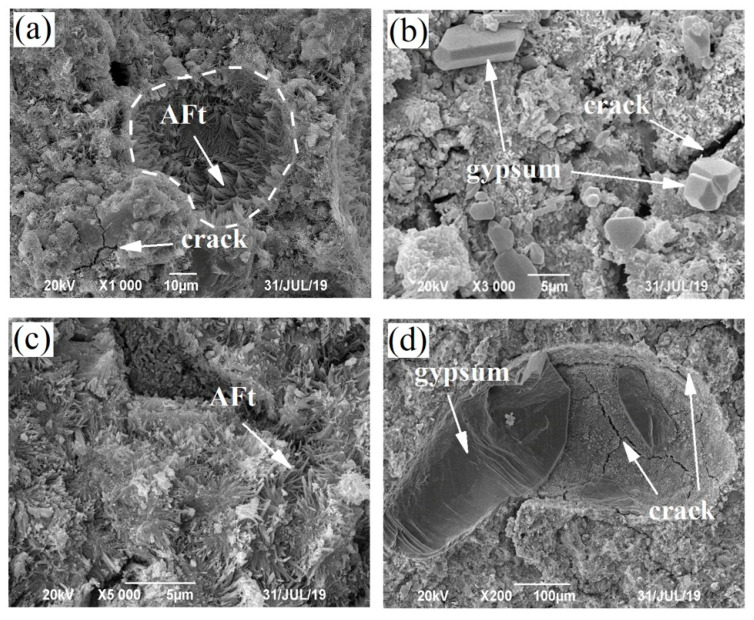
Partially immersed specimens for 80 days, SEM image of (**a**) and (**b**) PC specimen evaporation zone, (**c**) and (**d**) LS30 specimen immersion zone.

**Table 1 materials-15-08351-t001:** Chemical composition of cement and limestone (%).

	CaO	SiO_2_	Fe_2_O_3_	MgO	Al_2_O_3_	SO_3_	TiO_2_	K_2_O
Cement	62.68	19.62	2.96	1.89	4.37	2.06	0.24	0.71
Limestone	54.99	0.21	0.07	0.55	0.24	0.63	/	0.01

**Table 2 materials-15-08351-t002:** Mixing proportions of the cement pastes.

Number	Cement/%	Limestone/%	W/B
PC	100	0	0.55
LS10	90	10	
LS20	80	20	
LS30	70	30	

## Data Availability

The data presented in this study are available on request from the corresponding author.

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
