# Peer review of "The Damage Performance of Uncarbonated Limestone Cement Pastes Partially Exposed to Na2SO4 Solution"

_materials, 2022, doi:10.3390/ma15238351_

Round 1

Reviewer 1 Report

The manuscript with the title "The Damage Performance of Cement Pastes Partially Exposed to Na2SO4 Solution Avoiding Carbonation" and Manuscript ID: materials-2032101 by Cui et al. The work is appropriate for the journal, there are some observations that must be addressed before the manuscript can be accepted.

- Title needs to be improved.

- The abstract should be concise and specific and consequently should be revised.

- The abstract should clearly indicate the relevance of the work for international research.

- The authors should summarize the central core of knowledge that is the focus of the paper and better discuss the importance.

- The last part of the introduction should conclude the limitations of the previous studies and provide the main objectives and novelties of this study. You need to clearly address the knowledge gap and provide some meaningful phrases that your study can advance the knowledge and can fill in a knowledge gap that has not been considered yet.

- Research methods should be elaborated and justified.

- Describe the methods chronologically. This is very important to help the readers to replicate your results. Please cite previous research studies where necessary.

- The figures have not been appropriately explained as well. The readers cannot perceive the main points. Please describe the critical points and trends in the figures.

- The authors should discuss the potential cause of results and not only describe that it happens. In addition, the results should be discussed more deeply in respect to other studies.

- The authors must work harder in the explanation of the results since in work, they found very interesting data that must be discussed in greater depth.

- Selected references are quite old, which from the one point of view is good, since the authors cited necessary references to define a research problem, while from the other hand, lack of recent references may indicate an insufficiently performed literature review.

- The authors must work harder in the explanation of the results since in work, they found very interesting data that must be discussed in greater depth.

- Grammar and syntax must be improved.

Reviewer 2 Report

The introduction could be supplemented with information on the impact of sodium sulfate on the durability of concrete structures after certain periods of time, the loss of mechanical strength.

Hydration mechanisms of cement paste, some hydrates are sensitive to the presence of sulfate ions in solution.

After what age of hardening were the samples removed from the solution?

the limiting parameter of adsorption, how is the C/S ratio in relation to the free sulfate content?

What amount of sulfate was physically adsorbed?

Does the pH value of the exposure solution influence the leaching?

Did you notice if longitudinal or transverse cracks appeared after a certain period of exposure to the sodium sulfate solution? Have you checked the cement paste if it has lost its cohesion and if it has deteriorated after a longer period?

Do you have photos, images of the degradation of the cement paste and after other periods of exposure to the sulfate solution?

Reviewer 3 Report

This paper focuses on the damage performance of cement pastes partially exposed to Na2SO4 Solution avoiding carbonation. I think this is an important result for understanding the pore structures effect on sulfate attack. However, several points as indicated below need to be addressed by authors to improve the quality of the articles.

1. p.2, Figure1

Please add captions to ①-⑤ in the figure.

2. p.6, Figure5

The vertical scales of the bar graphs in both figures should be aligned.

3. pp.6-7, Figure6

Please add captions of abbreviations such as “E” and “G”, etc.

4. p.8, Line252-254

Is the content of ettringite and gypsum based on the results of Figure.8? If so, I think the results of Figure8 should be explained here.

5. p.8, Figure8

Is the legend incorrect? According to the sentences (Line265-267), the blue bar is probably ettringite. Please confirm the data.

6. p.8, Line265

“L30” replace by “LS30”.

7. p.8, Line266

Does the outer layer mean “E1”? If so, please add E1 with it.

8. p.9, Figure9, Line284

Which area does the circular pore boundary in Figure9(a) mean? Please specify in the figure.

Round 2

Reviewer 1 Report

It can be accept now, as the authors have revised the manuscript.